# Theta sequences as eligibility traces: A biological solution to credit assignment

**Tom M George**
Sainsbury Wellcome Centre for Neural Circuits and Behaviour
University College London, London, UK
`tom.george.20@ucl.ac.uk`

## Abstract

Credit assignment problems, for example policy evaluation in RL, often require bootstrapping prediction errors through preceding states *or* maintaining temporally extended memory traces; solutions which are unfavourable or implausible for biological networks of neurons. We propose theta sequences – chains of neural activity during theta oscillations in the hippocampus, thought to represent rapid playthroughs of awake behaviour – as a solution. By analysing and simulating a model for theta sequences we show they compress behaviour such that existing but short $\mathsf{O}(10)$ ms neuronal memory traces are effectively extended allowing for bootstrap-free credit assignment without long memory traces, equivalent to the use of eligibility traces in TD($\lambda$).

## 1 Introduction

When one decodes position, $x_E$, from the hippocampus (HPC) of a rodent it sweeps from behind to in front of the *true* position, $x_T$, once every theta cycle (a strong 5-10 Hz neural oscillation). So-called "theta sequences" (Foster & Wilson, 2007) don't make sense if the only goal of HPC is to accurately encode self-location at all times, they likely service some other objective (Drieu & Zugaro, 2019). Building off a body of literature linking fast hippocampal phenomena to learning and RL (Mehta et al., 2000; Bono et al., 2021; George et al., 2022b), here we demonstrate that theta sequences accelerate learning analagous to how eligibility traces (ETs) accelerate policy evaluation in RL.

Policy evaluation with temporal difference (TD) learning permits two kinds of solutions: prediction errors can be bootstrapped through preceeding states one-by-one (TD(0)) or temporally extended ETs can be maintained so credit can be assigned to states directly (Monte-Carlo, aka. TD(1)). These approaches are unified by the TD($\lambda$) algorithm (Sutton, 1988) (Appx. A).

Learning with long ETs, TD($\lambda > 0$), is typically faster, and therefore desirable, but biologically implausible since individual neurons have no trivial way to maintain ETs over timescales significantly longer than the membrane time constant $\mathsf{O}(10-50)$ ms. Perhaps theta sequences provide a solution to this problem: starting behind and moving in front of the animal rapidly within each cycle, the series of states observed within a sequence is an exact temporal compression of the states encountered on behavioural timescales (Fig. 1a). In this regime the short neuronal ETs are magnified by the same compression factor and long ETs are indirectly achieved (Appx. C).

We derive the relationship between TD($\lambda$) and theta sequences and empirically test it on a simple policy evaluation task (Fig. 1b) by comparing artificial agents implementing TD($\lambda$) with varing $\lambda$'s (Fig. 1c) to biological agents with short eligibility traces, TD($\lambda \approx 0$), undergoing theta sequences of varying velocity (Fig. 1d).

## 2 Results

Temporal difference learning using bioplausibly short ETs, $\tau_z = 10$ ms, on theta sequences is algorithmically equivalent to learning with long ETs $\tau_z^{\text{eff}}$ without theta sequences (see Appendix).

The effective compression is given by the ratio of the sequence velocity to the true agent velocity

$$\tau_z^{\text{eff}} = \frac{|\dot{x}_E|}{|\dot{x}_T|}\tau_z. \tag{1}$$

Agents move at a constant velocity of $v_T = 10$ cm s$^{-1}$ around a 2 m track upon which a small reward is located, whilst learning the value function (Fig. 1b). Increasing theta sequence velocity accelerates learning for the biological agent similar to how increasing the ET timescale accelerates learning for the artificial agent (Fig. 1cd, top panel). When sequence velocity is low, learning resembles heavily bootstrapped TD(0) with the value function slowly creeping back from the reward site over time. When sequence velocity is high, learning resembles TD(1) with credit appropriately assigned to all states simultaneously (Fig. 1cd, bottom panel). Biologically realistic sequence velocities (2 - 10 ms$^{-1}$ (Wikenheiser & Redish, 2015)) match the range in our model where there is a sharp change from TD(0)-like to TD(1)-like learning regimes.

Small errors can be observed in biological learning (Fig. 1d doesn't converge to 1 for slower sequence speeds) due to, we suspect, 'loop effects' (Appx. D) occuring when the sequence discontinuously resets once per theta cycle. These loop effect are not catastrophic for learning. Despite these effects we actually find learning on theta sequences is overall *less* noisy (compare value estimates in Fig. 1c and d) probably because, where the artifical agent can visit a location once per lap, theta sequences can traverse a location multiple times, smoothing learning. Learning with very fast sequences outpaces the artifical equivalent, probably because a single sweep (the very first) can explore the entire environment whereas the sequence-less artifical agent must wait until at least one lap for it to have observed all states. In reality sweeps this fast are not observed in the brain.

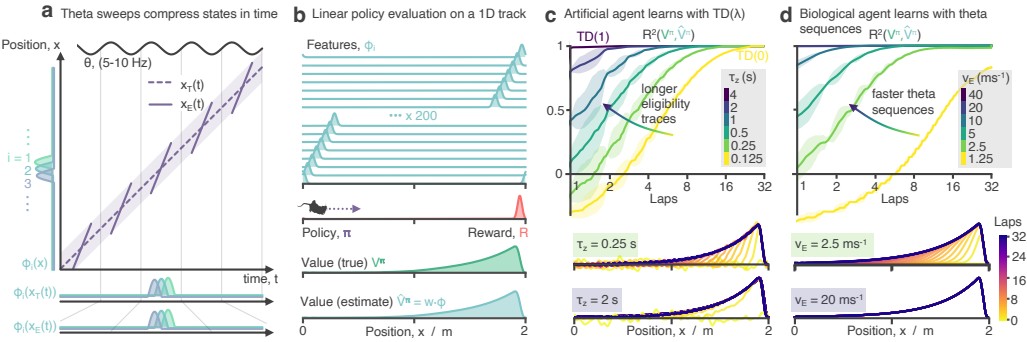

Figure 1: **a** Theta sequences: $x_E(t)$ (encoded position), sweeps from behind to infront of $x_T(t)$ (true position), compressing spatial inputs therefore indirectly extending memory traces. **b** A policy evaluation task on a periodic 1D track. The value function is approximated as a linear sum of Gaussian basis features. **c** An artificial agent learns with TD($\lambda$). (Top) Learning curves showing R$^2$ between true and estimated value functions for increasingly long eligibility traces (increasing $\lambda$). (Bottom) Evolution of the value estimate over learning for two opposing regimes: short eligibility traces (lots of bootstrapping) and long eligibility traces (no/little bootstrapping) **d** As in panel c but a biological agent with short eligibility traces 10 ms learns with theta sequences of increasing velocity. Sequence velocities are chosen to match eligibility timescales in panel c according to our simple theory.

## 3 CONCLUSIONS

Theta sequences provide a viable mechanism by which biological networks of neurons can perform long-term credit assigment without resorting to slow bootstrapping nor maintaining implausible long memory traces. Increasing sequence velocity is equivalent to increasing $\lambda$ – using longer ETs – in TD($\lambda$). Interestingly, in the brain theta power correlates with environmental uncertainty (Cavanagh et al., 2011) as well as periods of learning (Joensen et al., 2023) and sequence velocity depends on an animal's proximity to reward (Wikenheiser & Redish, 2015); based on the results shown here we conjecture that top-down processes may actively control theta sequence speeds in order to accelerate or slow down learning depending on local conditions.

## 4 CODE

Code reproducing results in this paper can be found at `https://github.com/TomGeorge1234/ThetaSequencesAreEligibilityTraces`

## 5 ACKNOWLEDGEMENTS

I thank Caswell Barry, William de Cothi, Claudia Clopath and Kimberly Stachenfeld for their support and feedback on this manuscript and the ideas within.

### URM STATEMENT

The author acknowledges that they meet at least one URM criteria of ICLR 2023 Tiny Papers Track.

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

## A  POLICY EVALUATION

In our model an agent at position $x_T(t)$ moves at a constant speed, $\dot{x}_T(t) = v_T = 10$ cm s$^{-1}$ from left to right around a periodic 1D track of circumference 2 m. A small reward density, $R(x)$, is centred at the far end of the track (Fig. 1b). The goal of the agent is the learn the value function for the current policy, defined as the discounted integral of future reward

$$V^\pi(x) = \int_t^\infty e^{-\frac{t'-t}{\tau}} R(x(t'))dt' \mid x(t) = x \tag{2}$$

over a discount time horizon $\tau = 4$ s. This is done using a linear approximation, a weighted sum of independent features

$$\hat{V}^\pi(x) = \sum_i w_i \phi_i(x) \approx V^\pi(x). \tag{3}$$

Famously, this problem can be solved with a temporal difference learning rule

$$\dot{w}_i(t) = \eta \delta(t) z_i(t) \tag{4}$$

where $\delta(t)$ is the (temporally continuous) TD error

$$\delta(t) = R(t) + \frac{d\hat{V}^\pi(t)}{dt} - \frac{\hat{V}(t)}{\tau} \tag{5}$$

and $z_i(t)$ is the *eligibility trace* of the $i^{\text{th}}$ feature

$$z_i(t) = \int_{-\infty}^t e^{\frac{t'-t}{\tau_z}} \phi_i(x(t'))dt'. \tag{6}$$

where $\tau_z \in [0, \tau]$ is the decay time scale of the eligibility trace.

The basis features are a set of 200 small Gaussian receptive fields ($\sigma = 2$ cm, 95.45% firing field therefore measures $4\sigma = 8$ cm), roughly analagous to place cells (O'Keefe & Dostrovsky, 1971) in the hippocampal formation, even spaced at 1 cm intervals along the track. Their small size means each features overlaps with approximately only its nearest meighbours. The reward density is another equally sized Gaussian at 1.95 m along the track.

We choose this simple policy evaluation task because it admits an analytical solution for $V^\pi(x)$. Since the policy is non-stochastic one can simply evaluate the integral in eqn. (2) accounting for the circular boundary conditions and compare this to the value estimate learnt by agents using temporal difference learning.

All simulations (agent trajectory, theta sweeps, neural activities and policy evaluation) were produced using the RatInABox simulation package (George et al., 2022a).

### A.1  RELATION TO DISCRETE RL AND TD($\lambda$)

It is more common to see the temporally-*discrete* formulation of policy evaluation with TD learning (nb. for a full discussion/derivation of continuous RL see Doya (2000)), summarised by

$$V^\pi(s_t) = \sum_{t'=t}^\infty \gamma^{t'-t} R(s_{t'}) \tag{7}$$

$$z_t = \sum_{t'=-\infty}^t (\lambda\gamma)^{t-t'} R(s_{t'}), \tag{8}$$

where $t$ is now a discrete integer state index. $\gamma$, the 'discount factor' determines over how many future states the agent cares about reward and can be compare this to $\tau$, the temporally continuous 'discount time horizon', determining how long (as a unit of time) into the future the agent cares about reward. $\lambda$ controls the decay-rate of the eligibility trace from $\lambda = 0$ (heavily bootstrapped regime) to $\lambda = 1$ (direct credit assignment, aka. online Monte-Carlo). This discrete formulation

is equivalent to the continuous one we use here in the integral limit of short timesteps where the following relationships become apparent

$$\gamma = e^{-\frac{dt}{\tau}} \tag{9}$$

$$\gamma\lambda = e^{-\frac{dt}{\tau_z}}. \tag{10}$$

Enabling us to link to two extremes of TD($\lambda$) as

$$TD(0) \Leftrightarrow \tau_z = 0 \tag{11}$$

$$TD(1) \Leftrightarrow \tau_z = \tau. \tag{12}$$

TD(0) (full-bootstrapping regime) occurs when the eligibility trace timescale falls to zero and TD(1) (Monte-carlo style learning) equates to when the eligibilty trace timescale matches the discount time horizon.

## B   THE ARTIFICIAL AGENT

The artificial agent learns according to the above TD learning rules and policy described in appendix A for a variety of eligibility trace timescales summarised in table 1:

| $\tau_z/s$ | 4 | 2 | 1 | 0.5 | 0.25 | 0.125 |
|---|---|---|---|---|---|---|
| $\tau/s$ | 4 | 4 | 4 | 4 | 4 | 4 |
| $\eta_{\text{opt}}$ | 0.4 | 0.5 | 0.6 | 0.8 | 1.1 | 1.3 |

Table 1: Learning parameters for the artificial agents.

Note the inclusion of two extremes: TD(1) ($\tau_z = \tau = 4$) and TD($\sim 0$) ($\tau_z = 0.125 \approx 0$). In order to be sure that small learning rates was not bottlenecking learning we optimised $\eta$ for each experiment by way of hyperparameter sweep (optimal value shown in table). When comparing the value estimate $\hat{V}^\pi(x)$ to the analytic value function $V^\pi(x)$ we use the coefficient of determination

$$R^2(V^\pi, \hat{V}^\pi) = 1 - \frac{\sum_x(\hat{V}^\pi(x) - V^\pi(x))^2}{\sum_x(V^\pi(x) - \langle V^\pi(x)\rangle_x)} \tag{13}$$

where the value estimate is first normalised to have the same maximum as $V^\pi(x)$ so, strictly, we are only comparing the shapes of the curves in Fig. 1cd (bottom panels).

The agent is allowed to explore and learn for a total of 640 s, exactly 32 laps, or until such a point that $R^2(V^\pi, \hat{V}^\pi)$ has been above 0.99 for a the entire previous lap, whichever comes first. Agents start from a random initial position $x_T(0) \sim \mathsf{U}(0\,\text{m}, 2\,\text{m})$. Plots/error bars show the average/std over 50 such experiments in the case of the artifical agent and 10 in the case of the biological agent.

## C   THE BIOLOGICAL AGENT

The biological agent differs from the artificial agent in two ways:

- **Short eligibility traces**: $\tau_z$ is fixed to a 0.01 s to emulate the biological contraint that neuronal memory times are short $\mathsf{O}(10)$ ms.
- **Theta sequences**: The firing rate of the features and the reward density are determined by the *encoded position* of the agent $x_E(t)$, not the true position $x_T(t)$. $x_E(t)$ sweeps from behind to in front of $x_T(t)$ in each theta cycle as described below.

Theta is modelled as a background oscillation of frequency $\nu_\theta = 1/T_\theta = 5$ Hz with a phase (used later) defined as

$$\phi_\theta(t) = \frac{t}{T_\theta} \mod 1 \tag{14}$$

During the middle fraction, $\beta = 0.75$ of each theta cycle $x_E(t)$ traverses symetrically from behind to in front of the agents true position at a speed of $v_E = v_T + v_S$ where $v_S$ is the speed of $x_E(t)$

in the reference frame of the true position. Outside there is no registered position and all firing rates are zero. Formulaically this can be stated as

$$x_E(t) = \begin{cases} x_T(t) + (\phi_\theta - 0.5)T_\theta v_S, & \text{if} \quad \frac{1-\beta}{2} < \phi_\theta(t) \le \frac{1+\beta}{2} \\ \text{None}, & \text{otherwise} \end{cases} \tag{15}$$

which determines the neural firing rates used for learning $\phi_i(x_E(t))$ and $R(x_E(t))$ where $\phi_i(\text{None}) = \mathbb{R}(\text{None}) := 0$

This brings us to the core hypothesis of this paper: **Since theta sequences traverse space faster than the real agent, the neural trajectory traverses the features faster than the real agent, compressing them. This compression means short eligibility traces, though remaining short, have more bang for their buck, effectively extending them**. The compression factor is

$$\kappa : \frac{v_E}{v_T} \implies \tau_z^{\text{eff}} = \kappa\tau_z. \tag{16}$$

Additionally, the same compression effect applies to the discount time horizon, $\tau$, such that, in uncompressed time coordinates it will have effectively increased,

$$\tau^{\text{eff}} = \kappa\tau. \tag{17}$$

so in order to learn a value function with (effective) discount time horizon of $\tau = 4$ s, $\tau$ must be decreased accordingly.

| $v_E/ms^{-1}$ | 40 | 20 | 10 | 5 | 2.5 | 1.25 |
|---|---|---|---|---|---|---|
| $\kappa$ | 400 | 200 | 100 | 50 | 25 | 12.5 |
| $\tau_z/s$ | 0.01 | 0.01 | 0.01 | 0.01 | 0.01 | 0.01 |
| $\tau$ / s | 0.01 | 0.02 | 0.04 | 0.08 | 0.16 | 0.32 |
| $\eta_{\text{opt}}$ | 8 | 8 | 2 | 2 | 2 | 0.75 |
| $\tau^{\text{eff}}$ | 4 | 4 | 4 | 4 | 4 | 4 |
| $\tau_z^{\text{eff}}/s$ | 4 | 2 | 1 | 0.5 | 0.25 | 0.125 |

Table 2: Learning parameters for the biological agents, Fig. 1d and their artificial equivalents.

Table 2 show the sweep velocities for six agents tested. These are carefully selected to match – according to our theory – the eligibility trace timescales of the six artifical agents. Hence the final two rows show the 'effective' behaviour, i.e. if the theory is correct, which artificial agent (no theta sequences and any choice of $\tau_z$) would this be equivalent to.

Learning only occurs *within* sequences ($\frac{1-\beta}{2} < \phi_\theta(t) \le \frac{1+\beta}{2}$). Outside this range (when there is no relevant data to learn from) learning is turned off ($\eta = 0$) reminiscent of the observation that hippocampal plasticity (LTP) oscillates significantly within each theta cycle (Hasselmo & Stern, 2014).

## D    LOOP EFFECTS

The results shown in Fig. 1d for the biological agent don't precisely converge to the value function for the slower sequences. We propose this may be due to 'loop-effects'. At the end of each theta cycle the sequence resets by discontinuously jumping back to a location behind the agent, Fig. 1a. This discontinuity could induce errors to grow within the value estimate: whereas the neural activity *during* the sequence can be seen as a sped-up replica of the true state trajectory, this discontinuity does not reflect any real transition statistics.

It is notable, therefore, that performance decay isn't catastrophic (all biological agents learn reasonable estimates of the value function) and is less pronounced for faster sequences, perhaps because the states at either end are further apart and interfere less. It is possible (but not tested) that the fraction of the cycle where there is no sweep ($1 - \beta$) allows existing short ETs to decay to zero essentially "forgetting" the jump transition and ameliorating the problem.

