# OpenReview forum: "Theta sequences as eligibility traces: A biological solution to credit assignment"
_ICLR.cc/2023/TinyPapers — Submitted to Tiny Papers @ ICLR 2023_

### Official Review · Reviewer_TvdM · 2023-03-24

**Confidence:** 3

**Summary Of Contributions:**

The authors present a method using theta sequences to solve the credit assignment problem in a reinforcement learning task. They motivate their approach from a neuroscience perspective and explain its enhanced biological plausibility compared to alternative approaches. They then test their approach compared to the alternative temporal difference approach on a simple reward learning task.

**Rating:**

High Impact (HI): a submission which meets the reviewing criteria and is predicted to make an impact on the field

**Strengths And Weaknesses:**

## Strengths
This is a very strong submission which meets all of the reviewing criteria and provides important neuroscientific hypotheses (specifically, what the role of theta sequences may be in the brain).

Clarity: The paper is very clearly written and is well-grounded in the relevant literature. There is one figure which is both visually appealing and effective.

Correctness: The results provided in the figure appear to support their claim that theta sequences accelerate learning.

Reproducibility: The authors provide extensive details of their approach in the appendix and will provide their code after publication.

## Weaknesses
The main weakness I see in this paper is that it's difficult to understand the method without reading the appendix. There is a lot of content here which may be better suited to a longer-form paper, and I leave it to the chair to decide whether Tiny Papers is an appropriate venue. If so, this could be a good submission to highlight in some way.

**Suggested Changes:**

Without reference to the appendix, it is difficult to understand exactly what theta sequences are (both in the brain and in this model). I understand that it's challenging to add more detail given the space constraints so this is mostly something to keep in mind for presenting this work. It would also help to see how the changing velocity of the theta sequence impacts the encoded position graph in Fig. 1a.

---

### Official Review · Reviewer_sMTd · 2023-03-30

**Confidence:** 2

**Summary Of Contributions:**

The paper studies the use of `theta sequences' and derives a relationship between theta sequences with TD(\lambda) used for policy evaluation tasks in reinforcement learning. An advantage with using Theta sequences is that these provide a viable mechanism by which natural neurons can perform credit assignment

**Rating:**

Clear, Correct, and Reproducible (CCR): a submission which meets the reviewing criteria

**Strengths And Weaknesses:**

The reviewer has very little familiarity with theta sequences and related topics used to build up hypotheses in the paper. As a result, the feedback included below is based on a limited understanding on the part of the reviewer

**Strengths**

- The paper provides a clear definition of the problem in the abstract itself. `Use theta sequences to solve credit assignment without bootstrapping and without long memory traces`.
- Figure 1(c) and 1(d) clearly provide evidence that support the above hypothesis. Specifically bootstrap-free credit assignment by biological neurons whose behavior is similar to TD(\lambda) or large \lambda
- The paper promises code that suggests that the work can be replicated. But I am not sure if other artifacts/data is needed for a full reproducibility

**Weaknesses**

- The paper appears to touch on neuroscience which makes it applicable to a narrower segment of ICLR community. This in itself is not a weakness but an opportunity for the paper to lay out a gentle introduction to allow a broader audience to appreciate the work.

- The paper refers to Figure 2 that is missing in both main paper as well as the Appendix. Perhaps this typo is due to mislabeling Figure 1 as Figure 2?


**Suggested Changes:**

Noted in **Weaknesses**

---

### Meta-Review · Area_Chair_4TKU · 2023-04-06

**Recommendation:** Invite to present (notable)
**Confidence:** 4

**Metareview:**

As agreed among reviewers, this is a strong paper with clearly-defined problem and well-presented evidence. It also meets the CCR criteria.

**Summary:**

This paper proposes to use theta sequences for policy evaluation in reinforcement learning.

**Reason For Not Giving A Higher Recommendation:**

NA

**Reason For Not Giving A Lower Recommendation:**

- the paper is well written
- the evidence is clearly presented

---

### Decision · Program_Chairs · 2023-04-08

Invite to present (notable)